# Neuroinflammation and Oxidative Stress in Individuals Affected by DiGeorge Syndrome

**DOI:** 10.3390/ijms24044242

**Published:** 2023-02-20

**Authors:** Michela Menghi, Ginevra Micangeli, Francesca Tarani, Carolina Putotto, Federica Pirro, Alessandro Mariani, Carla Petrella, Federica Pulvirenti, Bianca Cinicola, Fiorenza Colloridi, Luigi Tarani, Marco Fiore

**Affiliations:** 1Department of Maternal Infantile and Urological Sciences, Sapienza University of Rome, 00185 Rome, Italy; 2Department of Internal, Anesthesiologic and Cardiovascular Clinical Sciences, Sapienza University of Rome, 00185 Rome, Italy; 3Institute of Biochemistry and Cell Biology, IBBC—CNR, 000185 Rome, Italy; 4Regional Reference Centre for Primary Immune Deficiencies, Azienda Ospedaliera Universitaria Policlinico Umberto I, 00185 Rome, Italy

**Keywords:** ROS, rare disease, Del 22, genetic disorder, inflammation, 22q11.2 region, 10p

## Abstract

DiGeorge syndrome (DGS) is a rare genetic disease caused by microdeletions of the 22q11.2 region (DGS1). A haploinsufficiency at 10p level has been proposed also as a DGS cause (DGS2). Clinical manifestations are variable. The most frequent features are thymic hypoplasia or aplasia with consequent immune deficiency, cardiac malformations, hypoparathyroidism, facial and palatine abnormalities, variable degrees of cognitive impairment and psychiatric disorders. The specific aim of this descriptive report is to discuss the correlation between oxidative stress and neuroinflammation in DGS patients with microdeletions of the 22q11.2 region. The deleted chromosomic region maps various genes involved in mitochondrial metabolisms, such as DGCR8 and TXNRD2, that could lead to reactive oxygen species (ROS) increased production and antioxidant depletion. Furthermore, increased levels of ROS in mitochondria would lead to the destruction of the projection neurons in the cerebral cortex with consequent neurocognitive impairment. Finally, the increase in modified protein belonging to the family of sulfoxide compounds and hexoses, acting as inhibitors of the IV and V mitochondria complex, could result in direct ROS overproduction. Neuroinflammation in DGS individuals could be directly related to the development of the syndrome’s characteristic psychiatric and cognitive disorders. In patients with psychotic disorders, the most frequent psychiatric manifestation in DGS, Th-17, Th-1 and Th-2 cells are increased with consequent elevation of proinflammatory cytokine IL-6 and IL1β. In patients with anxiety disorders, both CD3 and CD4 are increased. Some patients with autism spectrum disorders (ASDs) have an augmented level of proinflammatory cytokines IL-12, IL-6 and IL-1β, while IFNγ and the anti-inflammatory cytokine IL-10 seem to be reduced. Other data proposed that altered synaptic plasticity could be directly involved in DGS cognitive disorders. In conclusion, the use of antioxidants for restoring mitochondrial functionality in DGS could be a useful tool to protect cortical connectivity and cognitive behavior.

## 1. Introduction

DiGeorge syndrome (DGS) or del22q11.2 syndrome is a genetic disease that involves microdeletions of the 22q11.2 (DGS1) [1] and/or 10p13/10p14 boundary (DGS2) [2,3]. Along with the well-defined clinical manifestations, this syndrome might involve changes in the regulatory mechanisms of oxidative stress and neuroinflammation as shown in animal models [4]. The aim of this descriptive report on the del22q11.2 syndrome, abbreviated from this point on as DGS, is to evaluate the studies in the literature concerning the mechanisms underlying these alterations.

## 2. DiGeorge Syndrome

DGS is a genetic syndrome caused by a microdeletion of the 22q11.2 region. The incidence is about 1 in every 4000/6000 live births, although the real prevalence could be higher due to the lower diagnosis rate in developing countries [5] (Figure 1).

The deleted 22q11.2 region encodes for 90 different genes, even if only some of the deleted genes appear to be directly related to the manifestations of the syndrome. In fact, one of the genes mostly correlated with the typical anomalies of the syndrome is TBX1, belonging to the transcription factors of the T-box family [6]. This gene plays a key role in the genesis of mesoderm, endoderm and pharyngeal ectoderm. TBX1 loss involves anomalies in the correct genesis of the thymus, pharynx and mandibular region [6]. Furthermore, TBX1 is involved in the development of the right ventricle and the arterial cone, which could explain the cardiological features frequently found in DGS. Along with heart abnormalities, TBX is also involved in cerebral microvascular system formation, which could explain the DGS patient susceptibility to developing psychiatric pathologies [6,7].

Another typically deleted gene is DGCR8, encoding the DGCR8 microprocessor subunit, a double-stranded RNA protein that mediates the synthesis of several miRNAs [8]. Lacking DGCR8 alters the expression of genes adjacent to those of the deleted region, favoring the psychiatric manifestations associated with the syndrome [9] as also shown in animal models [10].

The DGS signs and symptoms are so varied that different combinations of its descriptions were once considered as independent conditions. These early classifications include also Shprintzen syndrome, DiGeorge sequence/syndrome, velocardiofacial syndrome, conotruncal anomaly face syndrome and Sedlackova syndrome [9,11,12,13,14]. Sometimes immunodeficiency or immune manifestations are not pronounced, which is generally required for the diagnosis of DiGeorge syndrome, even in patients where velocardiofacial symptoms are present [5,15,16]. All are now considered to be presentations of a single syndrome.

Indeed, the DGS clinical presentation is heterogeneous [1,7,17,18,19,20]. The decline in quality of life is strongly related to congenital cardiac defects, psychiatric disorders and palatal anomalies. Recurrent infections and autoimmune disorders (thrombocytopenia being the most common manifestation) due to immunodeficiency and immune-dysregulation caused by thymic dysfunction and hypoparathyroidism/neonatal hypocalcemia related to parathyroid abnormal development consistently contribute to the clinical phenotype [1]. Renal, vertebral and airway structural anomalies, hypotonia, growth and neurodevelopmental delay, microcephaly and hearing alterations are also present [5] (Figure 2).

Important features can be recapitulated using the mnemonic **CATCH-22** to indicate 22q11.2DS, with the 22 suggesting that the chromosomal aberration is located on the 22nd chromosome [21]:

Cardiac abnormality (commonly interrupted aortic arch, truncus arteriosus and tetralogy of Fallot)Abnormal faciesThymic aplasia or hypoplasiaCleft palateHypocalcemia/hypoparathyroidism early in life

DGS individuals may show many possible outcomes, fluctuating in number of associated outcomes and from the mild to the very severe. Symptoms demonstrated to be common comprise the following [1,7,17,18,19,20]:-Palatal abnormalities (50%), particularly velopharyngeal incompetence, submucosal cleft palate and cleft palate; characteristic facial features (present in the majority of Caucasian individuals) including hypertelorism;-Cyanosis (bluish skin due to poor circulation of oxygen-rich blood);-Congenital heart disease (40% of individuals), particularly conotruncal malformations (interrupted aortic arch (50%), pulmonary atresia, persistent truncus arteriosus (34%), tetralogy of Fallot and ventricular septal defect);-Hearing loss (both conductive and sensorineural) (hearing loss with craniofacial syndromes);-Hypocalcemia (50%) (due to hypoparathyroidism);-Significant feeding problems (30%);-Learning difficulties (90%), including cognitive deficits and attention deficit disorders;-Laryngo-tracheo-esophageal anomalies;-Growth hormone deficiency;-Renal anomalies (37%);-Psychiatric disorders;-Autoimmune disorders;-Immunodeficiency present in about 75% of patients, related to thymic aplasia or hypoplasia determining alterations in both humoral and cell-mediated immune responses;-Immune disorders due to reduced T cell numbers;-Schizophrenia develops in 25–30% by adulthood;-Seizures (with or without hypocalcemia);-Skeletal abnormalities.

Facial features of DGS are quite typical and are represented by hypertelorism, narrow palpebral rims, coloboma, small auricles implanted down and rotated anteriorly, the helix folded, the nose with broad root and root hypoplasia, smallmouth and micrognathia which usually decreases with age [22] (Figure 3).

Psychiatric disorders may have a very negative impact on the quality of life of these patients and include anxiety, attention disorders, autism spectrum disorders and schizophrenia; the latter seems to affect between 25% and 40% of subjects with DGS [23].

In addition, 20% of patients affected have unilateral facial paralysis due to hypoplasia of the depressor muscle in the corner of the mouth (DAOM) that causes an asymmetry of the lower lip, evident especially during crying [5].

The diagnosis is genetic by performing array-CGH. This can be effectuated during pregnancy or after birth in case of clinical suspicion [24]. Life expectation of these patients is reduced due to psychiatric and cardiological comorbidities related to the syndrome [7,23].

## 3. Oxidative Stress in DiGeorge Syndrome

Oxidative stress plays an important role in the development of different signs and symptoms of various genetic syndromes [25]. The correlation between the imbalance of the redox state and the clinical DGS manifestations has not yet been established, but important findings have been revealed, as shown in Table 1 [26]. In particular, in 2019, Fernandez et al. used a mouse model of DiGeorge/22q11 deletion syndrome to describe biological mechanisms of neuronal under-connectivity, which reflects reduced growth of dendrite, axon and synapse [4]. It seems that connections between and within regions of the cerebral cortex are the basis for complex behaviors [27]. It has been proposed that the under-connectivity of association cortices underlies behavioral deficits in several neurodevelopmental disorders, including DGS [28].

Observing functional images, it has been demonstrated that association cortices are under-connected in the DGS [32]. Fernandez et al. demonstrated that under-connectivity occurs due to inefficient reactive oxygen species (ROS) catabolism and mitochondria-associated oxidative stress in layer 2/3 projection neurons (responsible for association cortico-cortical connections), while layer 5/6 neurons were unaffected [4].

In addition, the reduced dosage of the gene Txnrd2, a 22q11 gene essential for reactive oxygen species clearance in brain mitochondria, is responsible for the disruption of layer 2/3 projection neurons’ antioxidant defense and cellular consequences.

They also demonstrated that the use of antioxidants or the re-expression of Txnrd2 improved quantitative cellular deficits associated with LgDel layer 2/3 PN-selective under-connectivity. Accordingly, the oxidative stress could be considered a therapeutic target in neurodevelopmental disorders, as in DGS, due to the connection between the antioxidant restoration of mitochondrial integrity, cortical connectivity and cognitive behavior [4].

The relationship between DGS and alterations in the function of mitochondria has been also studied by Napoli et al. in 2015 [8]. Under high-oxidative-stress conditions, the mtDNA copy number per cell and mtDNA deletions are usually increased. Napoli et al. found that the mtDNA copy number and mtDNA deletions were both increased in the peripheral blood monocytic cells in DGS patients. Thus, several plasma metabolites were also indicative of increased oxidative stress. Especially, they observed an increased flux through the pentose phosphate shunt and products of oxidatively modified proteins (such as Met sulfoxide from Met, indole-3-lactate from Trp and aminomalonate) and hexoses (5-hydroxymethyl-2-furoic acid, hexuronic acid) [8].

In addition, they found that the elevation of 2HG in plasma could reflect a significantly higher intracellular (in particular intramitochondrial) concentration of 2HG. Accumulation of this metabolite has the potential to inhibit Complex IV, Complex V and creatine kinase and also increase oxidative stress, leading to a secondary block of the tricarboxylic acid (TCA) cycle.

This could result in a higher ratio of [NADH]/[NAD] and [FADH2]/[FAD], followed by the production of NADPH through the mitochondrial nicotinamide nucleotide transhydrogenase to sustain isocitrate dehydrogenase 2 activity and the reductive carboxylation of glutamine to citrate [8]. Furthermore, they pointed out that 9 of the 30 genes involved in DGS have the potential of disrupting mitochondrial metabolism. They especially focused on the TXNRD2 gene.

TXRND2 is one of the six 22q11 mitochondrial genes, with PRODH and ZDHHC8, that have SNPs associated with schizophrenia [31,33]. TXNRD2 is elevated in schizophrenic-brain samples [34]. The TXNRD2 is a gene that encodes for a mitochondrial thioredoxin reductase. This thioredoxin reductase catalyzes the reduction of the active disulfide of thioredoxin 2 and other substrates, having an important role in antioxidant defense [35].

An altered dosage of one of the six 22q11 mitochondrial genes could influence mitochondria function causing changes in synaptic development or function that could provide to the increased susceptibility for psychopathology in DGS as schizophrenia [36,37]. Recently, the relationship between premature cellular senescence and mitochondrial oxidative stress has been investigated by silencing the gene of the critical region 8 of the DiGeorge syndrome (DGCR8), an essential component of miRNA biogenesis [26]. Indeed, the authors studied the role of microRNAs (miRNAs) in regulating cellular senescence in human mesenchymal stem cells, depleting miRNAs in the DGCR8. They found that the human mesenchymal stem cells with the DGCR8 knockdown showed critical proliferation defects and alterations associated with senescence, including increased levels of ROS. Transcriptomic analysis studies revealed that the antioxidant gene superoxide dismutase 2 (SOD2) was significantly downregulated in DGCR8 knockdown hMSCs. They also discovered that DGCR8 silencing in hMSCs caused hypermethylation in CpG islands upstream of SOD2. The treatment with 5-aza-2′-deoxycytidine reestablished SOD2 expression and ROS levels [26].

In conclusion, it is still not possible to describe a direct connection between oxidative stress and all the clinical manifestations of patients with 22q11DS deletion, but recent studies have pointed out that there could be a link between the psychiatric manifestations and alteration of crucial genes for the mitochondrial metabolism, leading to a reduction of antioxidant defenses. Further studies are needed to evaluate this connection and eventually the benefit of an antioxidant therapy.

## 4. Neuroinflammation

Pediatric neuroinflammation is a quite common condition for many pediatric syndromes such as Klinefelter and Down syndromes and in the Fetal Alcohol Spectrum Disorders [38,39,40,41,42]. Indeed, neuroinflammation is a condition related to inflammation of the central nervous system or spinal cord with activation of microglia. This inflammatory process can be induced by infections, traumatic injuries, toxic products or autoimmune mechanisms [43]. Microglia cells play a central role as they determine the innate immune response in the brain. This favors the production of proinflammatory cytokines, such as IL-1β, IL-6 and TNF-α, chemokines such as CCL2, CCL5 and CXCL1 and reactive oxygen species, defined ROS [42,44].

It should be noted that the central nervous system is an immunologically privileged site thanks to the presence of the blood-brain barrier, formed by endothelial cells and astrocytes [45]. This system protects the brain from peripheral inflammatory mechanisms, but any harmful insults that alter the permeability of the blood–brain barrier help to increase the innate immune response by promoting neuroinflammation [46]. Microglia cells make up about 20% of brain glial cells and act by protecting the brain from any inflammatory insults using the typical mechanisms of the innate immune response such as phagocytosis and the production of cytotoxic components [43,47]. These cells are endowed with high plasticity due to the rapidity of action they must exert to eliminate any harmful components in the brain [45].

After the inflammatory insult, the microglia favor the repair of damaged cells by recalling neurons and astrocytes to the damaged site [48]. Clearly, this mechanism of protection and repair of the nervous system requires a very precise balance which, if it is altered by inflammatory phenomena, will promote cell damage with consequent production of toxic substances for the brain [49].

Patients affected by DGS have a high incidence rate of psychiatric pathologies [9]. These appear to be directly related to neuroinflammation as shown in Table 2 [50]. About 60% of patients with DGS suffer from psychiatric disorders. The most frequent are psychotic disorders, autism spectrum disorders (ASD), anxiety disorders, attention deficit, depression and bipolar disorder [1,23,51,52]. It seems that 25–40% of patients with DGS are affected by psychotic disorders, with a 20–40 times higher risk than the general population [1,53].

Regarding patients with DGS, it is not possible to genetically trace the higher incidence of schizophrenia spectrum disorders (SSD), although both the catechol-O-methyltransferase (COMT) and the proline dehydrogenase (PRODH) gene map in the deleted region [54,55]. The first encodes a protein that has the function of methylating dopamine, whose loss, as happens in patients with del22q11.2, could cause cognitive and behavioral alterations; the second one encodes proline dehydrogenase which allows the correct elimination of proline whose accumulation, in large amounts in the brain, would favor the development of psychiatric disorders [54,55,56] (Table 2).
ijms-24-04242-t002_Table 2Table 2Relations between the neuropsychological disorders and markers of neuroinflammations in patients affected by DGS. ↑ indicates elevation. ↓ indicates decrease.Neuroinflammation in DiGeorge SyndromeClinical Manifestations: Markers of Neuroinflammation:Anxiety disorders↑ in CD3 and CD4 [52]Autism-related disorders↑ in IL-12, IL- 6, IL-1β, and IFNγ [57]↓ in IL-10 [57]Cognitive problems(data also from animal models)↑ sarcoendoplasmic reticulum calcium-ATPase type 2 (SERCA2) [30,58]Alterations of synaptic plasticity [30,58]Psychotic disorders↑ in IL-6 and IL1β [59]↑ in neutrophil to leukocyte ratio (NLR) [60]↓ in T-reg cells [59]

From an etiological point of view, the cause of these SSD clinical features is not completely known, but neuroinflammation seems to have a key role [61]. In fact, it has been shown that in patients affected by SSD there would be an abnormal activation of microglia with consequent production of proinflammatory cytokines, such as IL-6 and IL1β, and a simultaneous reduction of T cells. The T cell regulation would seem to reduce their activity to the detriment of T-helper 17, 1 and 2 cells, which would favor the inflammatory response [54,59].

An increase in Th17 cells could cause a reduction in hippocampal neurogenesis and a reduction in the synthesis of BDNF with consequent alteration of the blood–brain barrier (BBB) favoring neuroinflammation [47]. For example, in a 2018 study, it was found that in a sample of 34 patients with DGS, mean Th-17 cell values were higher than in 34 healthy controls, and in DGS patients, those with psychotic symptoms had higher rates of Th17 than those without psychotic symptoms, confirming the important role of T-helper proinflammation cells in the development of psychosis [59].

E. Mekori-Domachevsky et al. showed how in 13 patients suffering from DGS with psychotic symptoms, serum levels of IL-6 and IL-10 were significantly higher than in 36 controls with DGS without psychotic symptoms and 30 healthy controls [62]. In another more recent study, E. Mekori-Domachevsky et al., studying the same sample of patients as the previous work, also showed how the neutrophils to leukocytes ratio (NLR) was statistically increased in the group of DGS patients with psychotic symptoms [53,60]. The NLR is calculated as the neutrophil count divided by the lymphocyte count and correlates innate immunity with acquired immunity increasing in proinflammatory diseases [60]. Although these data could demonstrate the relationship between neuroinflammation and psychotic disorders, further studies are needed to better understand it.

As for ASD, it would seem that about 20–50% of patients with DGS suffer from this type of psychiatric disorder, significantly compromising their quality of life [23]. Hidding et al. in 2016 tried to demonstrate that there could be a correlation between the loss of COMT gene activity with a consequent increase in dopamine in the prefrontal region [63]. In their study, it was highlighted that the mutation of the COMT gene would determine an association with the increase in proline, related to the loss of the PRODH gene, causing more severe forms of autism [55,63].

The mutations of these two genes that are also involved in the etiopathogenesis of psychotic disorders would underline how the accumulation of proinflammatory metabolites would favor the development of psychiatric disorders in these patients [63,64]. As for the cytokine and proinflammatory component, we know from the general literature that healthy patients with ASD have greater activation of microglia as one of the first events in the neuroinflammatory process [65,66]. It is not still clear why there is greater activation of microglia in these patients, but it would involve the activation of the inflammatory cascade as assessed by the systematic review conducted by Liao X et al. [67].

In the context of patients with DGS, Ross H. et al. conducted the first study on the evaluation of neuroinflammation and the direct correlation with ASD [57]. They demonstrated that the proinflammatory cytokines IL-12, IL-6, IL-1β and INFγ were significantly increased in patients with severe forms of autism, while the increase in the anti-inflammatory IL-10 was a protective factor against autism [57]. Considering the increase in cytokines, the study reported how the neuroinflammation ultimately led to a reduction of Treg cells with anti-inflammatory activity, activating the cytokine cascade [5,57].

Neuroinflammation could also play an important role in the development of anxiety disorders, which affect about 40% of patients with DGS [1]. *Dou Y.* et al. studied a cohort of 159 patients with DGS, of which 123 suffered from psychiatric disorders; 80 patients of these 123 suffered from anxiety disorders [52]. The study found that CD3 and CD4 values were higher in patients with anxiety disorders alone than in unaffected controls [52]. These data validate the idea that the inflammatory response is directly related to the development of psychiatric pathologies.

It has been suggested that patients with del22q11.2DS might present alterations in synaptic plasticity [68]. Indeed, cognitive and psychiatric deficits are directly related to alterations of cortical and hippocampal plasticity. This idea was first demonstrated in a mouse model with del22q11.2DS studied by Earls L.R. et al. in 2009 where they assessed that enhanced neurotransmitter release in Df (16) 1/+ mice coincided with altered calcium kinetics in CA3 presynaptic terminals and upregulated sarcoendoplasmic reticulum calcium-ATPase type 2 (SERCA2) [58]. SERCA inhibitors rescued synaptic phenotypes of Df (16) 1/+ mice. This Ca dysregulation would favor the cognitive symptoms typical of these patients [30,68].

Astrocytes also play a key role in synaptic plasticity, as well as being directly involved in the phenomena of neuroinflammation [62]. The alterations in the correct activation of astrocytes in this process could be useful to explain some of the psychiatric and cognitive manifestations of these patients [23]. In fact, astrocytes regulate ion transport, the pH of the extracellular microenvironment, the microvascular system and the release of neuromediators such as gamma-aminobutyric acid (GABA) and glutamate and the removal of pro-inflammatory metabolites [69].

The mechanisms underlying the alteration of astrocytic function in patients with DGS would always be attributable to the gene alterations of COMT and PRODH. As previously described, the COMT gene and the PRODH gene are interconnected: the reduced expression of PRODH would lead to an increase in the activity of the COMT gene with consequent uncontrolled activation of dopaminergic receptors causing cognitive and psychiatric disorders [8,55].

Furthermore, it has been shown that the PRODH gene is highly expressed in the neonatal period when there is a rapid expansion of glial cells and astrocytes, a process directly dependent on mitochondrial activity [55,56]. The deletion of the del22q11.2 region would seem to involve mitochondrial alterations with possible reduced activity of the mediator PPARγ co-activator 1α (PGC-1α), which would cause a reduced astrocytic maturation and thus favor neuro-inflammatory phenomena and reduction of synaptic neuroplasticity, as studied by Zehnder T. et al. in 2021 in animal models [69].

For the other psychiatric disorders that often accompany DiGeorge syndrome, such as ADHD or depressive disorders, there are currently no studies in the literature able to correlate neuroinflammation with clinical manifestations [1,23].

## 5. Conclusions

Based on the available information, it emerges that some of the clinical manifestations of patients with DGS are directly related to alterations both in the mitochondrial mechanisms that regulate oxidative stress and in the mechanisms of neuroinflammation.

As for oxidative stress, there are no studies explaining the clinical relationship between DGS and oxidative stress, but there are interesting preliminary studies about the molecular and biological alterations related to them. In fact, the loss of the two genes DGCR8 and TXNRD2 may determine an increase in ROS production and, at the same time, antioxidant depletion.

In DGS mouse models showing alterations in mitochondrial function, it has been observed that a loss of projection neurons in layers 2 and 3 of the cortex is associated with significant cognitive impairments. Furthermore, it emerges that neuroinflammation is directly linked to the development of psychiatric symptoms, especially psychotic, anxious and those related to ASD. In fact, it is clear from the literature that the increase in proinflammatory cytokines, such as IL-1, IL-6 and TNF-α, and imbalances in T cells are directly responsible for the development of psychiatric disorders [70,71,72].

At the same time, neuroinflammation would alter the mechanisms underlying the correct development of the brain and synaptic plasticity, confirming its importance also in the development of cognitive disorders. Thus, the use of antioxidants such as the polyphenols extracted from plants, possessing also anti-inflammatory properties, could be a useful tool to counteract the deleterious DGS effects on the affected individuals as previously shown in humans and animals models for other pathologies [73,74,75,76,77].

## Figures and Tables

**Figure 1 ijms-24-04242-f001:**
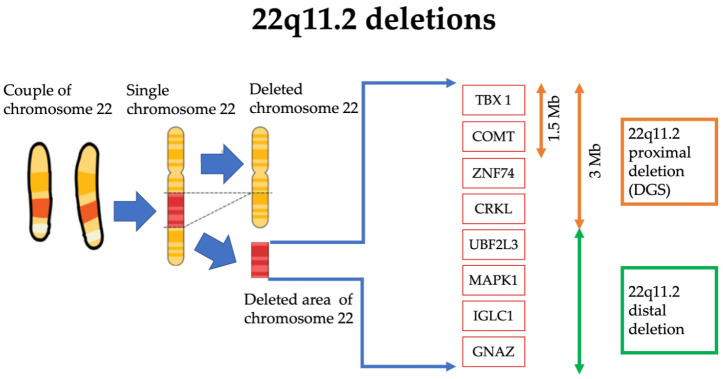
Schematic representation of the 22q11.2 deleted region of the chromosome 22 in DGS. The dimension of the deleted section (in red) is amplified for visual and graphical reasons.

**Figure 2 ijms-24-04242-f002:**
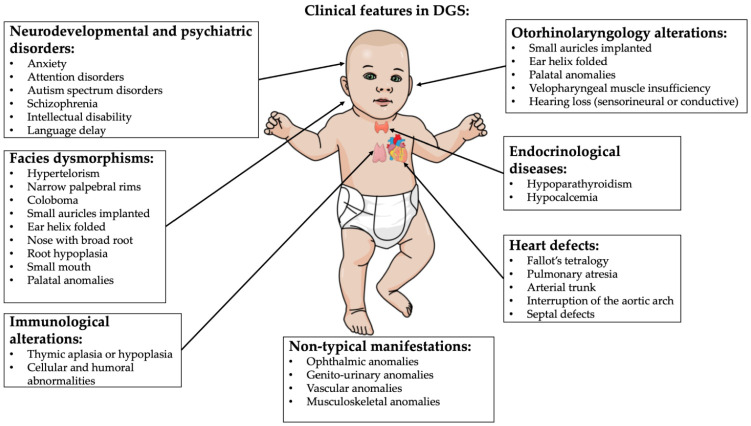
DGS typical clinical features. Image created by mindthegraph.com platform.

**Figure 3 ijms-24-04242-f003:**
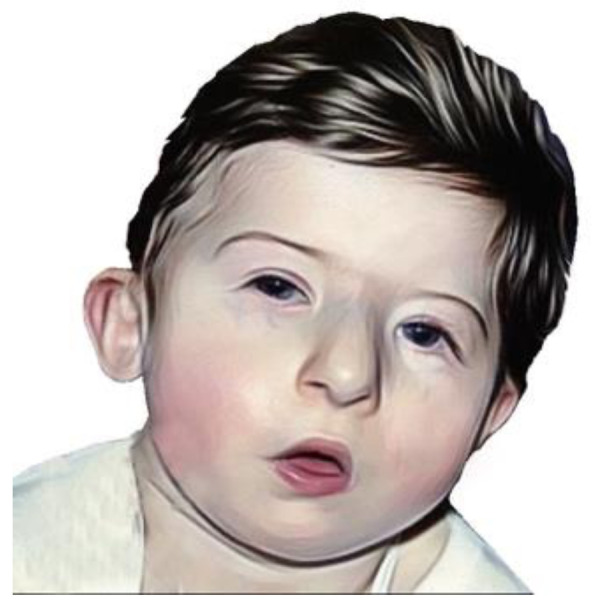
A DSG patient showing characteristic facial appearance, with tubular nose and carp-shaped mouth (redrawn from the original picture free of use under Creative Commons Attribution-Share Alike 3.0).

**Table 1 ijms-24-04242-t001:** Relations between genetic or enzymatic alterations and markers of oxidative stress in patients affected by DGS.

Oxidative Stress in DiGeorge Syndrome	
Genetic or Enzymatic Alterations:	Markers of Oxidative Stress:
Increase in mitochondria-associated oxidative stress in layer 2/3 projection neurons in mouse models	Disruption of layer 2/3 projection neurons related to association cortico-cortical connections [4]
Overproduction of modified proteins such as Met sulfoxide from Met, indole-3-lactate from Trp and aminomalonate Overproduction of hexoses such as 5-hydroxymethyl-2-furanoic acid and hexuronic acid in children	Inhibition of Complex IV, Complex V and creatine kinase [8]
Loss of DGCR8 gene in animal models	Inhibition of superoxide dismutase 2 (SOD2) [29,30] Increase in ROS [29,30] Accelerated senescence [29,30]
Loss of TXNRD2 gene in children	Reduction in antioxidant defense [8,31]

## Data Availability

Not applicable since this is a review paper.

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
