# Peer review of "Neuroinflammation and Oxidative Stress in Individuals Affected by DiGeorge Syndrome"

_ijms, 2023, doi:10.3390/ijms24044242_

Round 1
Reviewer 1 Report
Extensive line-by-line review below. In general, the review should be better structured to connect what has been described in MICE vs HUMANS, which in 22q11.2 deletion which in actual DiGeorge syndrome (see below). Some speculation on connections between genes and proteins is included, but particular findings in patients or model organisms are not very well explained.
Line 20: DGS is caused by other things as well, not only the typical 3Mb deletion.
Line 37: This conclusion is not justified by stated data at least in the abstract.
Line 45: not the deletion of the entire region, just part of it. Also, distal deletions in a non-typical region exist. No discussion or mention of chromosome 10 DGS2 in the entire manuscript.
Line 47: reference missing
Line 52: The deletion of 22q11.2 happens in 1:4000 births or so, DGS is a subset and is rarer. Diagnostic criteria of DGS is missing in the entire manuscript – 22q11.2DS is very variable, sometimes immunodeficiency or immune manifestations are not pronounced, which is generally required for the diagnosis of DiGeorge syndrome, even in patients where velocardiofacial symptoms are present. This should be discussed and taken into account with regard to studies discussed and included in the review. E.g. ESID diagnostic criteria.
Line 53: Figure 1 is not in the manuscript, or is labelled as Figure 3, showing the genes. Very unspecific picture. Selection of “genes related to the DGS” is arbitrary and not justified in the text. Justify why you selected these genes, improve the figure. Include distal deletions.
Line 66: DGCR8 in this case would not be mutated, but deleted.
Line 67: “some” miRNAs is too vague. Use several or list them.
Line 68-69: reference needed.
Line 75: Figure 2? The child is just a child with a heart, no characteristic organs affected are pictured. Use a better visual representation, link between boxes and picture.
Line 82: again, distinction between 22q11.2 deletion (syndrome) and DiGeorge syndrome.
Line 92: again labelled as Figure 2. What is the source of this photo, where is the patient consent listed? Is this taken from another paper, if so, is this copyright-OK?
Line 99: should be a new paragraph
Line 114: references 15 and 18 are identical. In Table 1 (and the rest of the text), distinction should be made for what studies were performed in mice and which in humans – this is very important throughout the text!
Line 150: I don’t understand why ref 24 is here?
Line 152: don’t use apparently
Line 159: why ref 29? Which gene of interest does it discuss?
Line 180: cite original works, not reviews. List which disorders neuroinflammation was described in, at least a couple.
Line 182: what is a weft injury?
Line 202: if you have subsection neuroinflammation in DGS but no other subsection in the Neuroinflammation section, is it really necessary? I would welcome section on neuroinflammation in Down’s, for example, or generally discussing particular findings in particular other diseases.
Line 207: cite original works, not reviews.
Line 211: SSD abbreviation not used in text before, only at line 221 which is LATER
Line 219: Table 2, ref 52 (Nain et al) do not show in their work a link between anxiety and elevated CD3 and CD4 counts. Double-check your references and claims! For cytokines or genes, list where were they evaluated (serum, CSF, etc). Table is missing list of abbreviations, e.g. NLR can mean NOD-like receptor, here it means neutrophil to leukocyte ratio. This can be misleading.
Line 225: if microglia activation leads to suppression of T cell response, discuss why in psychotic disorders in Table 2 you list Th1, 17, 2 as elevated. Also, you state that in SSD patients there is reduction of T cells, yet in Table 2 you list elevated CD3 and CD4 in patients with anxiety disorders…?
Line 260: ref 70 is the same as ref 54!
Line 263: Ref 71 has nothing to do with what is discussed in the sentence.
Line 269: ref 72 = ref 53…
Line 274: demonstrated in what studies? No neuroplasticity studies on HUMAN PATIENTS were cited in this review. Don’t use “patients” for mice.
Line 301: again, this was done in MICE, not PATIENTS. Strongly misleading.
Author Response
Reply to the criticisms raised by reviewer 1
Extensive line-by-line review below. In general, the review should be better structured to connect what has been described in MICE vs HUMANS, which in 22q11.2 deletion which in actual DiGeorge syndrome (see below). Some speculation on connections between genes and proteins is included, but particular findings in patients or model organisms are not very well explained.
Reply: We do thank the reviewer for the efforts she/he made in commenting on the work. We do believe that, according to the comments of the reviewers, the modifications we made in the revised text quite improved the scientific quality of the paper.
Line 20: DGS is caused by other things as well, not only the typical 3Mb deletion.
Reply: As requested, we rewrote the introductory and descriptive parts of the syndrome including also chromosome 10 DGS2 origin (pages 1 and 2 of the revised manuscript).
Line 37: This conclusion is not justified by stated data at least in the abstract.
Reply: as suggested we restated the sentence.
Line 45: not the deletion of the entire region, just part of it. Also, distal deletions in a non-typical region exist. No discussion or mention of chromosome 10 DGS2 in the entire manuscript.
Reply: As suggested we better described the general origin of the disorder including the chromosome 10 DGS2 issue (lines 17-21 and 43-49 of the revised manuscript).
Line 47: reference missing
Reply: as suggested, we rewrote the sentence.
Line 52: The deletion of 22q11.2 happens in 1:4000 births or so, DGS is a subset and is rarer. Diagnostic criteria of DGS is missing in the entire manuscript – 22q11.2DS is very variable, sometimes immunodeficiency or immune manifestations are not pronounced, which is generally required for the diagnosis of DiGeorge syndrome, even in patients where velocardiofacial symptoms are present. This should be discussed and taken into account with regard to studies discussed and included in the review.
Reply: As suggested, further info was provided. (pages 1, 2, 3, and 4 of the revised manuscript, text highlighted in light yellow).
Line 53: Figure 1 is not in the manuscript, or is labelled as Figure 3, showing the genes. Very unspecific picture. Selection of “genes related to the DGS” is arbitrary and not justified in the text. Justify why you selected these genes, improve the figure. Include distal deletions.
Reply: according to this comment Figure 1 was redrawn. We also updated Figure 2.
Line 66: DGCR8 in this case would not be mutated, but deleted.
Reply: we apologize for the typing mistake (line 68 of the revised manuscript).
Line 67: “some” miRNAs is too vague. Use several or list them.
Reply: as suggested, we modified the word (line 69 of the revised text).
Line 68-69: reference needed.
Reply: as requested, we included references (lines 71 and 72 of the revised manuscript).
Line 75: Figure 2? The child is just a child with a heart, no characteristic organs affected are pictured. Use a better visual representation, link between boxes and picture.
Reply: we apologize for the non-appropriate choice of the picture. As suggested, we have redrawn the figure.
Line 82: again, distinction between 22q11.2 deletion (syndrome) and DiGeorge syndrome.
Reply: As stated in the Intro, we discuss in this review the 22q11.2 deletion syndrome.
Line 92: again labelled as Figure 2. What is the source of this photo, where is the patient consent listed? Is this taken from another paper, if so, is this copyright-OK?
Reply: We did use a free-of-use figure under the Creative Commons Attribution-Share Alike 3.0. This kind of picture is free and may be used by anyone for any purpose. If you wish to use this kind of picture, you do not need to request permission as long as you follow any licensing requirements mentioned on this page. Anyway, we rewrote the caption providing more detailed information (lines 132-135 of the revised manuscript).
Line 99: should be a new paragraph
Reply: as requested, we created a new paragraph (line 142 of the revised text).
Line 114: references 15 and 18 are identical.
Reply: we do apologize for the typing mistake due to Mendeley malfunctioning. We did correct the numbers of the references.
In Table 1 (and the rest of the text), distinction should be made for what studies were performed in mice and which in humans – this is very important throughout the text!
Reply: We sincerely thank the reviewer for this comment. Appropriate modifications were carried out.
Line 150: I don’t understand why ref 24 is here?
Reply: As suggested ref 24 was deleted.
Line 152: don’t use apparently
Reply: as requested, the word was deleted.
Line 159: why ref 29? Which gene of interest does it discuss?
Reply: according to this comment we replaced the reference (line 202 of the revised manuscript).
Line 180: cite original works, not reviews. List which disorders neuroinflammation was described in, at least a couple.
Reply: We described 3 pediatric syndromes in these references (line 224 of the revised text).
Line 182: what is a weft injury?
Reply: We are sorry for the misunderstanding. We aimed at traumatic injuries (line 226 of the revised text).
Line 202: if you have subsection neuroinflammation in DGS but no other subsection in the Neuroinflammation section, is it really necessary? I would welcome section on neuroinflammation in Down’s, for example, or generally discussing particular findings in particular other diseases.
Reply: There is no need for a subsection. According to this comment, we deleted it.
Line 207: cite original works, not reviews.
Reply: further references were added (line 250 of the revised text).
Line 211: SSD abbreviation not used in text before, only at line 221 which is LATER
Reply: As observed, we made appropriate changes (line 254 of the revised manuscript).
Line 219: Table 2, ref 52 (Nain et al) do not show in their work a link between anxiety and elevated CD3 and CD4 counts. Double-check your references and claims! For cytokines or genes, list where were they evaluated (serum, CSF, etc). Table is missing list of abbreviations, e.g. NLR can mean NOD-like receptor, here it means neutrophil to leukocyte ratio. This can be misleading.
Reply: According to the comment of the reviewer table 2 has been updated.
Line 225: if microglia activation leads to suppression of T cell response, discuss why in psychotic disorders in Table 2 you list Th1, 17, 2 as elevated. Also, you state that in SSD patients there is reduction of T cells, yet in Table 2 you list elevated CD3 and CD4 in patients with anxiety disorders…?
Reply: Please see the previous reply.
Line 260: ref 70 is the same as ref 54!
Reply: Again, we do apologize for the Mendeley malfunctioning.
Line 263: Ref 71 has nothing to do with what is discussed in the sentence.
Reply: As suggested, ref 71 was deleted (now line 306).
Line 269: ref 72 = ref 53…
Reply: Again, we do apologize for the Mendeley malfunctioning.
Line 274: demonstrated in what studies? No neuroplasticity studies on HUMAN PATIENTS were cited in this review. Don’t use “patients” for mice.
Reply: As suggested, the sentence was rewritten (lines 317-324 of the revised text).
Line 301: again, this was done in MICE, not PATIENTS. Strongly misleading.
Reply: We apologize for the misleading info. Appropriate modifications were made (now line 344).
Reviewer 2 Report
The paper addresses one of the most interesting and emerging aspect of the DiGeoge Syndrome i.e the mechanisms underlaying the neurodevelopmental disorders. The historical nomenclature DiGeorge Syndrome in the title of the manuscript must be replaced with 22q11 deletion syndrome.The weakest part of the manuscript concerns the introductory and descriptive part of the syndrome which appears rather elementary .
Lines 70-74 :sentences to be adjusted.
Line 71: congenital cardiac: add defects; psychiatric: add disorders; palatal add anomalies. Line 71-72: instead of immunological abnormalities and hypocalcemia development : Recurrent infections and autoimmune disorders (being thrombocytopenia the most common manifestation) due to immunodeficiency and immunodysregulation caused by thymic dysfunction and hyoparathyroidis/neonatal hypocalcemia related to parathyroid abnormal development consistently contribute to the clinical phenotype.
Lines 72-73 : instead of laryngotracheal and esophageal abnormalities: airawy structural anomalies.
Line 73: eliminate trhombocytopenia
Line73: instead of growyh retardation growth and neurodevelopmental delay.
Concerning Fig.1: assembling clinical characteristics of patients with22q11.2 deletion is a challenge. However Fig. is quite scholastic and could be eliminated. In case the Authors wanted to keep it, they have to modify it.
Facies: facies dysmorphisms. Psychiatric alterations : neurodevelopmental and psychiatric disorders. Add in the related square: intellectual disability, language delay. Add a square about otolaryngological disorders. Immunological alterations: instead of alterations of T or B cells: cellular AND (not or) humoral abnormalities. Instead of edocrinological problems: endocrinological diseases. Add a square about"non typical manifestations": ophthalmic, genito-urinary, vascular, musculoskeletal anomalies.
The text contains several grammatical errors and english language must carefully revised.
some examples: Line 106: has not yet been established; Line118 :instead of arises occurs; Line 150: instead of studied focused on ; Line 160: senescence; Line 177: add "the benefit of" an antioxidant therapy; Line 194: modify "these cells are endowed with high plasticity due to the rapidity of action they must exert...
Author Response
Reply to the criticisms raised by reviewer 2
The paper addresses one of the most interesting and emerging aspect of the DiGeoge Syndrome i.e the mechanisms underlaying the neurodevelopmental disorders. The historical nomenclature DiGeorge Syndrome in the title of the manuscript must be replaced with 22q11 deletion syndrome..
Reply: We do thank the reviewer for the efforts she/he made in commenting on the work. We do believe that, according to the comments of the reviewers, the modifications we made in the revised text quite improved the scientific quality of the paper.
The weakest part of the manuscript concerns the introductory and descriptive part of the syndrome which appears rather elementary.
Reply: As suggested, the description of the syndrome has been updated by replacing also Figures 1 and 2. We did improve also the Tables (pages 1-4 of the revised manuscript, text highlighted in light yellow or light green).
Lines 70-74 :sentences to be adjusted.
Reply: According to the comment of the reviewer, we modified the sentence (now lines 81-88).
Line 71: congenital cardiac: add defects; psychiatric: add disorders; palatal add anomalies.
Reply: As suggested, we made the appropriate changes (now lines 81-88).
Line 71-72: instead of immunological abnormalities and hypocalcemia development : Recurrent infections and autoimmune disorders (being thrombocytopenia the most common manifestation) due to immunodeficiency and immunodysregulation caused by thymic dysfunction and hyoparathyroidis/neonatal hypocalcemia related to parathyroid abnormal development consistently contribute to the clinical phenotype.
Reply: As requested, the sentence was rewritten (now lines 81-88).
Lines 72-73 : instead of laryngotracheal and esophageal abnormalities: airawy structural anomalies.
Reply: as suggested, the sentence was rewritten (now lines 81-88).
Line 73: eliminate thrombocytopenia
Reply: as suggested, the word was deleted.
Line73: instead of growyh retardation growth and neurodevelopmental delay.
Reply: as suggested, the sentence was rewritten (now lines 81-88).
Concerning Fig.1: assembling clinical characteristics of patients with22q11.2 deletion is a challenge. However Fig. is quite scholastic and could be eliminated. In case the Authors wanted to keep it, they have to modify it.
Reply: According to the comments of the reviewers, Figure 1 has been redrawn.
Facies: facies dysmorphisms. Psychiatric alterations : neurodevelopmental and psychiatric disorders. Add in the related square: intellectual disability, language delay. Add a square about otolaryngological disorders. Immunological alterations: instead of alterations of T or B cells: cellular AND (not or) humoral abnormalities. Instead of edocrinological problems: endocrinological diseases. Add a square about"non typical manifestations": ophthalmic, genito-urinary, vascular, musculoskeletal anomalies.
Reply: We do thank the reviewer for the suggestions (included in the updated Figure 1).
The text contains several grammatical errors and english language must carefully revised.
some examples: Line 106: has not yet been established;
Reply: the sentence was corrected (now line 149).
Line118 :instead of arises occurs;
Reply: the sentence was corrected (line 161 of the revised text).
Line 150: instead of studied focused on ;
Reply: the sentence was corrected (line 193 of the revised manuscript).
Line 160: senescence;
Reply: the word was corrected (now line 203).
Line 177: add "the benefit of" an antioxidant therapy;
Reply: the sentence was corrected (now line 220).
Line 194: modify "these cells are endowed with high plasticity due to the rapidity of action they must exert...
Reply: the sentence was corrected (lines 238-240 of the revised text).